# Strategies for Mitigating Commercial Sensor Chip Variability with Experimental Design Controls

**DOI:** 10.3390/s23156703

**Published:** 2023-07-26

**Authors:** Eliza K. Hanson, Chien-Wei Wang, Lisa Minkoff, Rebecca J. Whelan

**Affiliations:** 1Department of Chemistry, University of Kansas, Lawrence, KS 66047, USA; e.hanson@ku.edu (E.K.H.); chienwei@ku.edu (C.W.); 2Department of Chemistry and Biochemistry, University of Notre Dame, Notre Dame, IN 46556, USA; minkoff.lisa95@gmail.com

**Keywords:** surface plasmon resonance (SPR), Ni^2+^-nitrilotriacetic acid (NTA), biosensor, immobilization, variability, Nicoya Life Sciences

## Abstract

Surface plasmon resonance (SPR) is a popular real-time technique for the measurement of binding affinity and kinetics, and bench-top instruments combine affordability and ease of use with other benefits of the technique. Biomolecular ligands labeled with the 6xHis tag can be immobilized onto sensing surfaces presenting the Ni^2+^-nitrilotriacetic acid (NTA) functional group. While Ni-NTA immobilization offers many advantages, including the ability to regenerate and reuse the sensors, its use can lead to signal variability between experimental replicates. We report here a study of factors contributing to this variability using the Nicoya OpenSPR as a model system and suggest ways to control for those factors, increasing the reproducibility and rigor of the data. Our model ligand/analyte pairs were two ovarian cancer biomarker proteins (MUC16 and HE4) and their corresponding monoclonal antibodies. We observed a broad range of non-specific binding across multiple NTA chips. Experiments run on the same chips had more consistent results in ligand immobilization and analyte binding than experiments run on different chips. Further assessment showed that different chips demonstrated different maximum immobilizations for the same concentration of injected protein. We also show a variety of relationships between ligand immobilization level and analyte response, which we attribute to steric crowding at high ligand concentrations. Using this calibration to inform experimental design, researchers can choose protein concentrations for immobilization corresponding to the linear range of analyte response. We are the first to demonstrate calibration and normalization as a strategy to increase reproducibility and data quality of these chips. Our study assesses a variety of factors affecting chip variability, addressing a gap in knowledge about commercially available sensor chips. Controlling for these factors in the process of experimental design will minimize variability in analyte signal when using these important sensing platforms.

## 1. Introduction

Surface plasmon resonance (SPR) was first used for biosensing in 1983 [1,2] and in the past forty years has become a powerful and widely used technique for the label-free measurement of binding affinity and kinetics. It can be applied to a variety of analytes, ranging from small molecules [3,4], nucleic acids [5,6], proteins [7,8], and antibodies [9] to whole cells [10]. In addition to determining binding kinetics, SPR has been used for diverse applications include epitope mapping [11,12], the detection and quantification of biomarkers [13,14], and studies of antibody inhibition [9,15].

The technique relies on the phenomenon of surface plasmon resonance. At a specific angle of incident light, the energy of the exciting photon couples with electrons in the sensing surface, which is typically a metal coating [2,16]. The resulting charge density wave is described as surface plasmons, and plasmon oscillation on the surface of the sensor generates an electric field [17]. Many commercial SPR instruments have relied on the attenuated total reflection method, which utilizes a high-refractive index prism in the Kretschmann geometry. The angle of incidence that gives rise to resonance is dependent on the refractive index of the metal surface and is affected by small changes caused by the immobilization of ligands to the surface of the chip or the binding of analytes [17,18]. Changes in binding on the surface are quantified by monitoring changes in the reflected light [17].

When the sensing surface contains nanoparticles at a size less than or equal to the wavelength of light in use, the free electrons have a collective oscillation defined as localized surface plasmons [19,20]. Localized surface plasmons have a maximum absorbance at a specific wavelength, which can be used to quantify the binding at the surface by monitoring the shift in the resonance wavelength. This variation on the technique is commonly called localized surface plasmon resonance (LSPR) [19,21]. Companies such as Nicoya have used LSPR to create affordable bench-top instrumentation [22].

There are a wide variety of strategies to immobilize ligands to the sensor chip surface, ranging from covalent coupling via carboxyl or thiol chemistry to affinity tag immobilization methods, including 6xHis-Ni-NTA linkage or streptavidin–biotin interactions. In particular, the 6xHis-Ni-NTA coupling strategy has numerous benefits that make it widely used. The small size of the 6xHis peptide tag results in minimal steric hindrance of the ligand compared to larger tags like glutathione S-transferase (GST), maltose binding protein (MBP), or streptavidin. It also offers the ability to control the orientation of immobilization [23] and the capacity to regenerate and reuse a sensor surface by utilizing imidazole or ethylenediaminetetraacetic acid (EDTA) to remove bound ligand from the surface. Between 2016 and 2022, 185 papers cited the use of a Nicoya OpenSPR instrument with one of their commercial sensor chips (sources: Nicoya website, PubMed, Web of Science, and Google Scholar.) Of these studies, 25.4% used the NTA sensor chemistry, demonstrating the importance of this immobilization strategy [22].

Our group has used the Nicoya OpenSPR-XT instrument and Nicoya NTA sensor chips to characterize the binding affinity of clinically used antibodies to recombinantly expressed repeat regions of biomarker protein MUC16 [24]. Our experience with the Nicoya NTA sensing platform indicated opportunities to improve rigor and reproducibility by identifying and correcting sources of signal variation. Though it is a popular immobilization strategy, little has been done to systematically characterize the sensor chips themselves. A 2007 study assessed protein desorption over the course of experiments on NTA chips [25], and one paper assesses the use of tris-NTA functional groups for immobilization instead of the more commonly used mono-NTA strategy [26]. In the current work, we demonstrate sources of variability (Figure 1) in NTA sensor chips using the Nicoya OpenSPR as an example. These sources of variability are common across other SPR systems. We offer suggestions for how to better design experiments to generate more reproducible and reliable data.

This study is a systematic characterization of the variability of commercially available NTA sensor chips for SPR and offers a more thorough understanding of the sources of sensor chip variability and strategies to combat them.

Techniques that rely on nanoparticle-sensing surfaces, such as LSPR or surface enhanced Raman spectroscopy (SERS), are popular detection methods. A growing area of research is developing new types of nanoparticle surfaces to improve sensing applications, such as the development of a SERS substrate based on diatom frustules from unicellular algae [27], upconversion nanoparticle surfaces [28], and bimetallic SERS biosensors [29]. As new nanoparticle-sensing surfaces are designed and existing systems grow in popularity, it is essential to ensure that experiments are critically designed to generate reproducible and trustworthy data, no matter what platform is used.

## 2. Materials and Methods

### 2.1. Recombinant Repeat Expression and Purification

The MUC16 sequence cited by O’Brien et al. [30] was sourced from NCBI (GenBank AF414442.2) and used to derive the nucleotide sequences of the nine tandem repeats used in this study (sequences found in Appendix A). The sequences were synthesized and cloned into pET14b vectors with XhoI and BamHI sites (plasmids purchased from GenScript, Piscataway, NJ, USA) to express individual tandem repeat proteins with N’ 6xHis-tagging. Plasmids were transformed into Shuffle T7 Express *E. coli* (New England Biolabs, Beverly, MA, USA), and clones were grown to exponential phase in Luria-Bertani (LB) broth (Thermo, Waltham, MA, USA) with 100 μg/mL of ampicillin at 30 °C. Protein expression was induced by adding 400 μM isopropyl β-D-1-thiogalactopyranoside (IPTG) for 4 h. Cells in phosphate-buffered saline (PBS) buffer with cOmpletet^TM^ protease inhibitor (Thermo) were harvested and lysed via freeze–thaw cycles. Ni^2+^-NTA beads (Thermo) were used to purify the 6xHis-tagged repeat proteins. Protein identities were verified via 6xHis antibody Western blots and mass spectrometry as described in our previous work [24] and as demonstrated in Appendix A.

### 2.2. Characterization of Tandem Repeat Protein–Antibody Binding Affinity

All samples were run on a Nicoya Lifesciences OpenSPR-XT 2-channel instrument (Nicoya Lifesciences, Kitchener, ON, USA) with NTA-functionalized sensors from the same vendor. PBS-T (PBS with added 0.05% Tween 20 from Fisher Scientific, Waltham, MA, USA) was used as immobilization and running buffers. Default experimental temperature was set to 20 °C for autosampler tray and sensor. Protein concentration was estimated before each experiment using a NanoDrop 2000 (ThermoFisher, Waltham, MA, USA) in Protein A_280_ mode (set to 1 Abs = 1 mg/mL). Protein was then diluted to 125 nM in PBS-T with the exception of repeat 6 (R6), which required a higher concentration (500 nM) to achieve acceptable immobilization levels. The surface was regenerated with two injections of 10 mM Glycine-HCl (pH 1.5) and one injection of 350 mM EDTA (TCI America, Portland, OR, USA). This regeneration method was used to clean the surface before the start of each experiment and after each ligand–analyte pair was injected. Regeneration was optimized during experimental design to ensure ligand was completely removed prior to next immobilization.

The surface was activated for immobilization using 40 mM NiCl_2_ (BeanTown Chemical, Hudson, NH, USA), and recombinant repeat was immobilized on the surface. Recombinant 6xHis-tagged streptavidin (His-SA, at a stock concentration of 1 mg/mL in 20 mM Tris-HCl pH 7.5, purchased from Fitzgerald Industries, Acton, MA, USA, My Biosource, San Diego, CA, USA, and Abcam, Waltham, MA, USA) was used as a blocking molecule at a concentration of 0.75 μM. Following blocking, the antibodies were injected, and the affinity was measured. M11 (epitope group B, Agilent, Santa Clara, CA, USA) was injected at stock concentration. OC125 (epitope group A, Sigma, St. Louis, MO, USA) was injected at a dilution of 1:200. M11-like (epitope group B, M61703 clone, Fitzgerald) and OC125-like (epitope group A, M61704 clone, Fitzgerald) were injected at a dilution of 1:2000. After each antibody injection, the surface was regenerated, resulting in four cycles of ligand immobilization followed by analyte injection for each experiment. Three replicates of each repeat were run, with an extra replicate for R5. Three different repeats were run per chip, with the order of repeat immobilized on the chip scrambled between chips. All reagent injections were done at a flow rate of 20 μL/min, except for Glycine-HCl (150 μL/min) and EDTA (100 μL/min). All injections had a dissociation time of 270 s. The average signals from Channel 1 (reference) and Channel 2 (ligand-immobilized) and the Corrected signal (Channel 1 subtracted from Channel 2) were calculated over the interval of 500–525 s at the end of the dissociation phase using TraceDrawer Analysis 1.9.2 (Ridgeview Instruments, Uppsala, Sweden). Because the experiment was designed as a semi-quantitative comparison of binding analysis and only used one concentration of each analyte, binding kinetics were not characterized. We used nine different sensor chips from two different lots to run the entire set of repeats in triplicate. Chips 1–5 were from lot SND0927, and chips 6–9 were from lot SNE0111.

### 2.3. Comparison of Inter- and Intra-Chip Variability

The characterization of binding affinity between monoclonal antibodies and repeat proteins (described above) used three replicates analyzed across three separate sensor chips. To compare reproducibility with replicates collected on the same chip, we collected additional replicates of R58. On the chip used to collect the third replicates of R58 and of two other repeats, we ran fourth and fifth replicates of R58, for a total of three replicates of R58 on different chips and three replicates on the same chip with one experiment common to both. To assess reproducibility of an additional repeat, three replicates of R9 were run on a single chip. Experimental parameters were held the same for additional replicates, except for the determination of R9 concentration, which used a bicinchoninic acid (BCA) concentration assay rather than NanoDrop.

### 2.4. Measurement of Maximum Sensor Immobilization

To establish the maximum protein immobilization on a chip, we injected a saturating concentration of HE4 (1750 nM, VWR Lifesciences, Radnor, PA, USA) followed by a lower concentration (250 nM). We first immobilized His-SA (0.75 μM) and regenerated the surface. Three cycles of high–low protein immobilization were performed per chip, with regeneration between sets. The corrected signal from both injections was added together, and the total immobilization for all three cycles was averaged. This experiment was run on three different sensor chips from two different lots. The first chip was from lot SND0927 and the second two were from lot SNE0111. HE4 concentration was measured using a BCA assay.

### 2.5. Calibrating the Relationship between Immobilization Level and Antibody Signal Magnitude

We performed a variety of calibrations with different protein–antibody pairs. A general experimental design for these calibrations is given in Figure 2. Four representative calibrations are reported in this paper: R11 and OC125, R11 and M11-like, R5 and OC125-like, and HE4 with anti-HE4-mAb. We immobilized a range of ligand concentrations to achieve different immobilization levels (Figure 2A) and held the antibody concentration constant from a pooled sample (Figure 2B), so that differences in antibody signal should correlate only with differences in immobilization level (Figure 2C).

The order of sample concentrations immobilized on the surface was randomized for all tests. For the tandem repeat proteins, we immobilized 25, 50, 100, 150, 200, and 250 nM protein, and antibody concentrations corresponded with those used for the tandem repeat binding affinity experiments. Flow rates, blocking, and dissociation time also matched the experimental conditions, with an additional EDTA injection during the regeneration cycle for a total of two Glycine-HCl injections and two EDTA injections. Tandem repeat protein concentration was estimated using the NanoDrop.

For the HE4 system, we immobilized 250, 500, 750, 1000, 1250, 1500 nM samples and held antibody concentration constant at 75 nM. The flow rates and blocking were the same as other experiments, and regeneration cycles used two Glycine-HCl injections and one EDTA injection. All signals were averaged over the 500–525 s interval, but the antibody samples had an extended dissociation time of 290 s instead of 270 s. HE4 and anti-HE4-mAb concentrations were measured with a BCA assay.

### 2.6. Normalization of HE4–Antibody Binding Affinity

A linear range of ligand immobilization level and antibody signal (R^2^ = 0.99) was chosen from the calibration and those HE4 concentrations were used in the following tests. We immobilized 250, 500, 750, and 1000 nM HE4 protein and injected antibody at a constant concentration of 75 nM. The regeneration conditions, flow rate, blocking steps, and extended dissociation time matched the conditions of the HE4 calibration experiments. Signal was averaged over the 500–525 s interval for both channels and the corrected output.

## 3. Results and Discussion

### 3.1. Non-Specific Binding of Antibodies across a Selection of Chips

We used four MUC16 antibodies to characterize the variability in non-specific binding (NSB) on different sensor chips. We averaged the signals resulting from injections of antibodies over a streptavidin-blocked sensing surface (Channel 1) from the four antibodies run in triplicate on each chip and collected the averages for all nine chips utilized in the experiment. Maxima and range data are shown in Table 1.

There was high variability in NSB observed on different chips, measured by changes in the reference channel signal, as shown in Figure 1. Across nine chips, M11-like had an NSB range of 498.32 RU, with the highest signal being 632.56 RU. M11 had a range of 452.30 RU with a maximum signal of 580.79 RU. While the other antibodies had lower maximum NSB signals and smaller ranges of NSB, they still demonstrate a broad range between different chips. OC125-like and OC125 have ranges of 303.81 and 199.34 RU, respectively.

The reference channel is intended to separate the effects of non-specific interactions from binding to the immobilized ligand, and thus we subtract it from the signal generated in the ligand-immobilized channel. We expect this subtraction to make the data more closely reflect the signal resulting from binding, but if the amount of NSB measured on different chips varies widely, it has the potential to affect how much signal we report for binding in the corrected signal after this subtraction. Because repeat protein is only immobilized in the ligand channel, it is possible to compare the reference channel signal for antibody injections across the three different experiments per chip. However, given that the experiments had only one replicate each for the binding of a given repeat with the four antibodies, and the magnitude of binding signal in channel 2 was very different for different repeats, a relative comparison of the ratio between signal in both channels cannot be made between different chips here. The range of total NSB observed from multiple runs on a single chip is smaller than the range observed across different chips, suggesting that replicates should be collected on a single chip whenever feasible.

### 3.2. Comparison of Inter- and Intra-Chip Variability

When we compared the experiments run on the same chip to the experiments run on different chips, we saw lower variability for both immobilization and antibody binding signals on the same chip versus different chips. The lowest immobilization variability was observed for R9 experiments on the same chip for which the RSD was 60.8 parts ppth. In contrast, the RSD of immobilization signal variability on different chips was 105.1 ppth. All RSD values are found in Table 2.

The lowest variability was observed for R9 samples, for which protein concentration was determined using BCA (rather than Nanodrop, which displayed day-to-day variability) and were run on the same chip (Figure 2A). The RSD values calculated for the binding signals (Figure 2B,C) show even more improvement between intra-chip experiments over inter-chip, with the RSDs for the same chip ranging from 42.78 to 95.62, whereas the RSDs for the analyte binding on different chips ranged from 252.24 to 449.17 (Table 2). We observe the same trends of lower variability for R58 experiments (data given in Appendix A and Table 2).

### 3.3. Establishing a Maximum Immobilization on Different Chips

We saw different maximum signals for immobilization of the same concentration of ligand on three sensor chips (Figure 3). The second injection of ligand ranged from 1.9 to 7.8% of the total signal, verifying that we had achieved sensor saturation. We see a difference of over 1000 RU between the highest and lowest maxima observed. Chip 1 was from a different sensor lot than Chips 2 and 3, and even between the two chips from the same lot, the maximum immobilization of protein differs by over 350 RU, demonstrating different immobilization efficiencies on different chips (Figure 3A). While not conclusive, the first His-SA injection on each chip demonstrated a similar pattern of variability (Figure 3B).

Though it lacked the same second injection to verify sensor saturation, a comparison of His-SA immobilization in Channel 1 throughout the tandem repeat binding affinity experiments offers a useful insight into maximum possible immobilizations on a chip, as all injections reflect the same concentration of His-SA immobilized (Figure 4). There is a difference of over 1300 RU between the highest and lowest maxima observed for His-SA protein in the reference channel across nine different chips, once again demonstrating that we cannot expect the same immobilization level for a given concentration of protein when immobilized onto different chips.

It is also useful to consider the above chips in terms of lot variability. The first five chips were from one lot and display much higher variability in their average immobilization level and the spread of measurements, with an RSD of 170.2 and SEM values for each chip ranging from 118.8 to 393.4. However, the last four chips were from a separate lot and display a much more consistent average and lower overall variability when considered either in terms of individual chips or as a whole lot. This second set had an RSD of 61.4, and the SEM values for the individual chips range from 36.5 to 209.0. For circumstances in which all experiments cannot be done on a single chip, there appears to still be a benefit to doing experiments on only chips derived from the same lot.

Also supporting the reproducibility of performing tests on the same chip, Nicoya has demonstrated immobilizing His-SA on an NTA chip up to 27 cycles, with only 2.9% variability [31].

### 3.4. Chip Thickness

In addition to differences in the performance of the sensor chips with regard to the functionalized nanoparticle sensing surface, we also observed variability in the thickness of the glass sensor chips. Using digital calipers, we measured 88 sensor chips (20 from known lots, 68 from unknown lots) and saw a range of 0.1 mm between the thickest chip and the thinnest (see Table 3, Appendix A). The docking of the sensor holder (how far forward the sensor stage is moved to seal to the flow cell) is sensitive to chip thickness and can affect the collection of data if too loose or too tight. Because the docking of the instrument had been optimized with a thicker glass chip, when we attempted to use a thinner chip, a difference of 0.08 mm in the thickness of the chips resulted in an incomplete docking seal and leaking so significant that it was not possible to run the experiment without adjustment. In addition to the chemical behavior of functionalized sensor chip surfaces, it is important to consider the physical dimensions of the chip and the possible effects they may have on the data collection of an experiment. Due to the nature of the experimental design, it is not possible to isolate the effects of chip thickness on data quality or reproducibility beyond the observed leaking that necessitated adjustment to perform the characterization. However, this is a valuable future investigation to undertake in order to better understand the effects of chip thickness on experimental design.

### 3.5. Calibrating the Relationship between Antibody Signal and Immobilization Level

We have seen that different chips achieve different possible immobilization maxima, and that the chips with more consistent immobilizations also had more consistent antibody-binding responses. We performed calibration experiments to determine a more exact relationship between the two parameters (see Figure 2) and observed several different relationships between immobilization and binding signal. For some ligand–analyte pairs, we saw a linear region where the immobilization level and antibody signal were positively correlated, followed by a leveling off in the antibody signal as the immobilization level increased (Figure 5A). We also saw ligand–analyte pairs in which there was a negative correlation between higher immobilization levels and antibody signal (Figure 5B) and those in which a significant drop in antibody signal occurred after a certain immobilization level (Figure 5C).

Different ligands have different maximum immobilization signals and ranges where analyte binding signal correlates positively with immobilization level. However, ligands that are similar (such as the tandem repeats) appear to converge on a similar linear positive correlation range, even if the binding past saturating immobilization levels has much more variation. For a set of ligands with similar properties, this would likely enable calibrating with a few ligand–analyte pairs to determine the best immobilization range instead of characterizing every single pair across the full immobilization range. It is important to understand the relationship between the immobilization level and analyte signal to ensure that the binding is being measured in the range where immobilization and analyte binding signal are positively correlated. Attempts to normalize a set of experiments without this initial calibration did not lead to more consistency in the data, and the variety of possible responses observed during the example calibrations sheds light on why this variability persisted. Using an immobilization level within the linear range is recommended because if immobilization level is too high and the surface is oversaturated, steric hindrance can obscure potential binding. By calibrating before performing affinity characterizations, the investigator can choose a ligand immobilization level that allows for the accurate measurement of binding and enables the normalization of the data for analysis.

### 3.6. Normalizing Binding Affinity Data to Account for Immobilization Variability

We used the calibration approach described in the previous section to characterize the relationship between HE4 immobilization and subsequent binding of its antibody (Figure 5D). We then immobilized HE4 at concentrations within the positive linear range. This allowed us to normalize the antibody-binding signal to the HE4 immobilization level, and we demonstrated consistent antibody binding despite different immobilization levels. The RSD for the corrected signals is 409.3 but drops to 159.1 after normalization. In addition, despite seeing lower overall HE4 immobilization as we collected each replicate, normalization was able to mitigate the effect of this immobilization drop (Figure 6).

## 4. Conclusions

The present work demonstrates a variety of sources of possible variability when performing experiments using NTA chips, and we suggest corresponding controls that researchers can use to minimize the observed variation. By running experiments on the same chip, the effect of variable non-specific binding and different immobilization capacities can be minimized. It is also critical to calibrate the relationship between immobilization and binding signal of a given ligand–analyte pair, because calibration removes variability from possible different immobilization levels and allows for more consistent data when binding signal drops over time. Calibration also helps to ensure that tests accurately reflect any possible analyte binding and that potential binding has not been obscured by a ligand immobilization level that is too high or too low. Future work will assess more quantitative binding analysis methods, including studying how chip variability affects the fitting of kinetic parameters and increasing the number of replicates performed to make violin plots and to better understand sensor chip variability on a broad scale. In addition, there are insights to be gained about the physical properties of the chip by characterizing the surface morphology of the nanoparticles and quantitatively assessing how the thickness of the sensor chips affects data quality and reproducibility. Identifying sources of variability in NTA-based sensing experiments offers the opportunity to correct those sources of variation, improving reproducibility and yielding a more accurate measurement of ligand–analyte binding.

## Data Availability

Data sharing not applicable.

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
