# Peer review of "Strategies for Mitigating Commercial Sensor Chip Variability with Experimental Design Controls"

_sensors, 2023, doi:10.3390/s23156703_

Round 1

Reviewer 1 Report

In this manuscript, the authors reported variability of Ni2+-nitrilotriacetic acid-functionalized sensor chips for surface plasmon resonance. This work seems to be useful for this field. However, the following problems should be addressed before further consideration of publication:

1. The title is suggested to be brief and concise. A scheme can be created after the Introduction section to better demonstrate the contents.

2. The abstract should be revised to address the development and novelty of this work, especially the superiority or enhancement when compared with other advances.

3. In the Introduction, references should be enriched to contain advances in the related fields. The advantages of SPR-based biosensing should be added including: 10.1002/EXP.20220072; 10.1016/j.apsusc.2020.148374; 10.1002/EXP.20210216.

4. Figure 5 is confusing, maybe some other styles are suitable for better demonstration.

5. All the figures need to be revised with consistent layout/size to improve the readability.

6. How to obtain the ligand-analyte calibration? The result diversity and significance in various stages should be explained in detail.

7. What do you think can further improve the performance, and is it feasible for future practical applications?

Reviewer 2 Report

This research work he demonstrates a variety of sources of possible variability when per- 339 forming experiments using NTA chips, and authors suggest corresponding controls that researchers can use to minimize the observed variation. Given the nature of LSPR measurement (high sensitivity) and the large variation for the testing results, my concern is that in some cases the sample size (n=3 or 4) might be not sufficient for the conclusion. Hence, in the future study, authors may increase the sample size and make violin plots for statistical analysis to give deeper insights into the use of the Nicoya OpenSPR-XT instrument and Nicoya NTA sensor chips. I suggest to accept the publication after the minor revision to address the following questions.

11)      At line 132, authors mentioned that 9 different sensor chips were used from 2 different lots to run the test. So authors should clearly state which sensor chips are from the same lot.

22)      For figure 4, the readings of sensor chips 1-4 seem to have statistical difference (significantly higher) with that of sensor chip 5-9. Is this statistical difference attributed to the lot difference?

33)      Authors pointed out that the physical dimensions of the sensor chips might affect the data collection of an experiment (line 288). Did author investigate the relationship between the thickness of the 9 different sensor chips with their experimental reading?

44)      To calibrate the relationship between immobilization level and antibody signal magnitude (Fig. 6), how many tests have been replicated (n=?)? If n3, where is the standard deviation for each point in Fig. 6?

Round 2

Reviewer 1 Report

I have checked all the revisions.

Reviewer 2 Report

I suggest publishing the manuscript as it is.